# Viral metagenome characterization reveals species-specific virome profiles in Triatominae populations from the southern United States

Insan Habib[1]☙, Juan Camilo Hernandez-Valencia[1]☙, Jana Martinu[1], Eva Novakova🆔[1,2]*

**1** University of South Bohemia, Faculty of Science, Ceske Budejovice, Czech Republic, **2** Biology Centre of the Czech Academy of Sciences, Institute of Parasitology, Ceske Budejovice, Czech Republic

☙ These authors contributed equally to this work.
* novake01@prf.jcu.cz

## Abstract

Kissing bugs (Triatominae) are hematophagous insects and the principal vectors of *Trypanosoma cruzi*, the causative agent of Chagas disease. While their bacterial microbiomes have received considerable attention, the diversity of viruses associated with these insects remains poorly understood. To address this gap, we investigated the metavirome of five Triatominae species from the southern United States (*Triatoma rubida, T. sanguisuga, T. gerstaeckeri, T. indictiva,* and *Hospesneotomae protracta*), sampled in Texas, New Mexico, and Arizona. We sequenced 23 samples, including abdomen, gut and reproductive tissues from 13 field-collected individuals and assembled 41 viral operational taxonomic units (vOTUs), 40 of which are novel and together constitute 13 viral families, including *Chuviridae, Arenaviridae, Orthomyxoviridae, Partitiviridae, Solemoviridae, Circoviridae, Rhabdoviridae, Microviridae, Xinmoviridae, Astroviridae, Narnaviridae, Tombusviridae,* and the order Elliovirales. The vOTUs composition and abundance analysis examined variables including species, sex, tissue type, blood meal, and *T. cruzi* infection status, showing that metavirome diversity varied significantly among Triatominae species. Our findings demonstrate a species-specific metavirome and the presence of virus taxa linked to insects, plants, and vertebrates, highlighting the complex ecological interactions between viruses and triatomines. This study uncovers a diverse and largely novel set of metaviromes within North American Triatominae, providing a foundation for future research on virus-vector interactions.

## Author summary

Kissing bugs are blood-feeding insects best known for transmitting *Trypanosoma cruzi*, the parasite that causes Chagas disease. Like most of other organisms, kissing bugs carry not only bacteria but also viruses as part of their natural

**Data availability statement:** All relevant data are within the manuscript and its Supporting information files. Raw RNA-seq reads have been deposited in the NCBI Sequence Read Archive (SRA) under BioProject accession number PRJNA1279500. Viral genome assemblies have been deposited in GenBank under accession numbers PX139058 - PX139102.

**Funding:** This work was supported by the Czech Science Foundation (grant number 21-10185M to EN). The funders had no role in study design, data collection and analysis, decision to publish, or preparation of the manuscript.

**Competing interests:** The authors have declared that no competing interests exist.

microbial communities. These viruses, often harmless and sometimes potentially beneficial, can influence how insects develop, respond to infections, and interact with the pathogens they transmit. Yet, compared to bacterial microbiomes, the viral communities of kissing bugs have remained almost completely unexplored. In this study, we analyzed metaviromes present in digestive and reproductive tissues of five species of kissing bugs collected across the southern United States. Using high-throughput sequencing, we discovered 41 distinct viral OTUs nearly all of them previously unknown. These represent a wide range of viral families, including those typically associated with insects, plants, and vertebrates. We also found that each kissing bug species carries its own unique metavirome. By uncovering this hidden virome diversity, our work provides a foundation for understanding how viruses might shape the biology, immunity, and transmission capacity of these important vectors.

## Introduction

Metagenomic and metatranscriptomic surveys in recent years have vastly expanded our understanding of the viromes associated with blood-feeding arthropods. These studies reveal that hematophagous vectors harbor a surprising diversity of viruses, ranging from well-known arboviruses to numerous insect-specific viruses (ISVs) that do not infect vertebrates [1]. While mosquitoes and ticks have received the most attention of vector virome research [2–5], kissing bugs (Triatominae) and their associated viruses have remained largely understudied. Despite the well-established role of Triatominae as vectors of *Trypanosoma cruzi*, the etiological agent of Chagas disease, current knowledge of their virome diversity, distribution, ecological dynamics, and host-virome interactions remains limited [6]. Until recently, only eight viruses had been described from Triatominae: the dicistrovirus Triatoma virus (TrV) and seven Rhodnius prolixus virus strains (RpV1-7). TrV, a non-enveloped positive-sense single-stranded RNA virus from the *Dicistroviridae* family, was first identified in *Triatoma infestans* populations in Argentina [7,8]. Subsequent studies demonstrated that TrV is the only known triatomine viral pathogen [9–12]. Due to its association with high nymphal mortality, delayed development, and impaired oogenesis, TrV has been proposed as a potential biological control agent for Chagas disease vectors [8,13]. In contrast, the phenotypic effects of the RpV1-7 viruses remain unknown. Discovered through transcriptome sequencing of *Rhodnius prolixus* tissues [14], these viruses are evolutionarily distinct from TrV and belong to the families *Iflaviridae* (RpV1, RpV2), *Permutotetraviridae* (RpV3, RpV4, RpV7), and *Solemoviridae* (RpV5, RpV6). Notably, both TrV and RpVs appear to be vertically transmitted to progeny [8,14], suggesting persistent infections within triatomine populations.

The first virome oriented surveys in Triatominae were conducted by Brito et al. [14] and Bourke et al. [15], who used RNA-seq data from public repositories to uncover a broader diversity of viruses associated with kissing bugs. Mining the datasets from

122 Triatominae specimens (eight species from six countries), the later study identified six distinct viruses, including the previously known RpV4-6 and widespread detection of TrV in nearly half of all screened individuals [15]. This work notably expanded TrV's known host range, reporting its presence for the first time in *Triatoma brasiliensis* and *Mepraia* species. Additionally, two novel viruses were detected in kissing bugs: Meccus longipennis virus 1 and Drosophila melanogaster Nora virus. The former was identified in multiple *T. infestans* populations across Argentina, Brazil, Chile, and Peru, while the latter (a picorna-like virus previously known from fruit flies) was found in *T. infestans* in Argentina. These findings emphasize the vast, yet underutilized, potential of publicly available datasets for identifying both known and novel viruses, while also revealing significant gaps in our understanding of Triatominae-associated viromes.

The present metatranscriptomic study focuses on five Triatominae species native to the southern United States and provides a comprehensive analysis of the associated viral metagenomes. By characterizing viral communities in both gut and reproductive tissues of *Triatoma rubida*, *Triatoma gerstaeckeri*, *Hospesneotomae protracta*, *Triatoma sanguisuga*, and *Triatoma indictiva*, we aim to gain insight into tissue-specific viral diversity and transmission potential. In addition, we investigate how ecological and biological factors, such as *T. cruzi* infection status and the identity of recent blood meals, influence virome composition and structure across individuals and species. Identifying novel viruses associated with these medically relevant insects may reveal candidates for biological control, as seen with TrV adversely affecting triatomine development and survival.

## Materials and methods

### Sample set

The sample set was designed to include digestive and reproductive tracts from 13 individuals of fifth instars, males, and females representing five Triatominae species sampled in Arizona, New Mexico, and Texas in 2022. The individuals retrieved by active searches through vertebrate nests and burrows were morphologically determined to species, pre-served in RNA*later* solution and stored in -20°C before dissection. For two females, only whole abdomens (tissue sample extension "ABD") could be processed because the abdominal incision made to allow RNA*later* infiltration prevented reliable separation of individual tissues. For the remaining 11 individuals (5 females, 4 males, and 2 L5s), reproductive tissues and digestive tracts (excluding the crop) were dissected under a Nikon SMZ745T stereomicroscope and preserved in liquid nitrogen (S1 Table, tissue sample extension "GONADS" and "GUT").

### Total DNA/RNA extraction, Next-generation sequencing, and blood meal analysis

DNA and RNA templates from 24 tissue samples were isolated with AllPrep DNA/RNA Micro Kit (Qiagen) following the manufacturer´s protocol. The quality of RNA was checked with Agilent 2100 Bioanalyzer (Agilent Technologies), and the sample concentration was measured using Qubit RNA High Sensitivity Assay Kit (Thermo Fisher Scientific). The samples passing QC (n = 23, S1 Table) were used for cDNA libraries construction including ribosomal RNA (rRNA) depletion, and sequenced as 150 bp long PE reads on Illumina NovaSeq 6000 by Novogene (United Kingdom). To identify the dominant bloodmeal source, analysis was conducted with 12 DNA isolates derived from gut tissues (S2 Table). We used vertebrate universal primers, amplifying 215 bp long fragment of the 12S rRNA gene: L1085 (5'-CCCAAACTGGGATTAGATACCC-3') and H1259 (5'-GTTTGCTGAAGATGGCGGTA-3') (Kitano et al. 2007). PCR products were purified and sent for Sanger sequencing. Identity of the sequences was assessed based on the first BLASTN hits against the core_nt database.

### Raw data processing, assembly, and viral metagenome identification

Quality control of 23 raw sequence datasets was performed using the Trimmomatic (v0.39) tool with default parameters to remove the low-quality reads. rRNA contamination was removed using ILLUMINACLIP by matching reads against a custom rRNA reference file compiled from identified overrepresented sequences [16]. Clean reads were used as input

and *de novo* assembled into contigs using MetaSPAdes (v3.15.2) (K-mers = 21, 33, 55; other parameters were kept at default) [17]. All the assembled contigs were aligned to the protein version (v26.0) of the Reference Viral Database (RVDB) [18] using the DIAMOND (v v0.9.14.115) with BLASTX algorithm using the --very-sensitive option, E-value of 1e-3 and the default parameters [19]. Additionally, to detect additional potential RNA-dependent RNA Polymerase (RdRp) from RNA viruses, publicly available HMM profiles of different RdRp of eukaryotic RNA virus families were obtained from Pfam database v36.0 (Mononega- and Chuviridae-like: Mononeg_RNA_pol [PF00946], Reoviridae-like: RdRP_5 [PF07925], Flaviviridae-like: Flavi_NS5 [PF00972], Bunyavirales-like: Bunya_RdRp [PF04196], Narnaviridae-like: Mito-vir_RNA_pol [PF05919], Nidovirales-like: RdRP_1 [PF00680], Tymovirales-like and Hepe-Virga-like: RdRP_2 [PF00978], Tombusviridae-like and Nodaviridae-like: RdRP_3 [PF00998], Toti-, Luteo-, and Sobemoviridae-like: RdRP_4 [PF02123], Viral_RdRp_C [PF17501], and Birnaviridae-like: Birna_RdRp [PF04197]), available through InterPro [20,21]. From all the retrieved profile HMMs a combined HMM profile database was built using *hmmbuild* from the HMMER suite [22]. The contigs were translated to six possible open reading frames (ORFs) obtained from the online VirtualRibosome tool (https://services.healthtech.dtu.dk/services/VirtualRibosome-2.0/) [23], and the amino acid (aa) sequences were used as an input for *hmm*search. To avoid false positives, the contigs with virus hits in RVDB and HMM were aligned against the NCBI's non-redundant (nr) protein database via NCBI-BLAST with an E-value of 1e-5 and the default parameters [24]. Further-more, the non-redundant contigs greater than 300 bp (nt) were selected for functional and structural annotation, and phylogenetic analyses. The virus sequences were taxonomically assigned following the guidelines for sequence-based taxonomy provided by the International Committee on Taxonomy of Viruses (ICTV) [25]. The newly discovered virus sequences were named based on the genus or subfamily and the Triatominae species, or the abbreviation "spp." if found in multiple species, followed by the standard virus taxon according to their taxonomic classification.

## Molecular confirmation of Triatominae species and their phylogenetic background for the virome analyses

Molecular confirmation of the involved species, especially *T. sanguisuga* and *T. indictiva* was performed by recovering 8 mitochondrial genes, including, Cytochrome C Oxidase I (COXI), Cytochrome C Oxidase II (COXII), Cytochrome C Oxidase III (COXIII), NADH dehydrogenase subunit 1 (ND1), NADH dehydrogenase subunit 4 (ND4), NADH dehydrogenase subunit 5 (ND5), cytochrome b (cytb) and 16S ribosomal RNA (16S rRNA) sequences, from each meta-assembly. The gene sequences were aligned separately using MAFFT algorithm implemented in Geneious Prime v.2022.0.2 (http://www.geneious.com/) and then concatenated into the final matrix containing 9032 nucleotides. The phylogenetic analysis was conducted with a maximum likelihood approach using PhyML. Bootstrap analysis was run with 100 iterations.

### *Trypanosoma cruzi* meta-assembly screening

To investigate viral metagenome diversity in the context of *Trypanosoma cruzi* infection, we explored our data to iden-tify infected and uninfected *Triatoma* and *Hospesneotomae* individuals. All the assembled contigs were first subjected to BLASTN searches against the custom database created from all available *T. cruzi* genomes from TriTryDB (https://tritrypdb.org/) [26]. Identity of all contigs producing an alignment of 200 bp and longer in the first search was confirmed by BLASTN against NCBI's nt database (S1 Table).

## Phylogenetic analysis and annotation of viral sequences

For structural and functional annotation of viral sequences, the amino acid sequence of identified ORFs was queried against the Protein Data Bank (PDB) database using HHpred method [27]. For the phylogenetic analysis of the assem-bled virus sequences, the amino acid sequences of the RdRp were mainly used since they are highly conserved in nature. Where available, we also used the nucleocapsid, and capsid sequences. The datasets were assembled from all BLASTP hits with more than 30% identity found in the nr database. The source organism and geographic metadata for all the hits

were retrieved using the NCBI Entrez-direct tool (v19.4) [28] with a custom bash script and added to the header of each FASTA file. To reconstruct phylogenetic relationships among identified virus sequences, all the datasets were aligned using MAFFT under automatic settings [29]. Aligned sequences were trimmed using trimAl v1.4 using the *-gappyout* algorithm to remove poorly aligned positions [30]. Maximum-likelihood trees were inferred in IQ-TREEE2, where *ModelFinder Plus* with partition merging (MFP+MERGE) selected the optimal amino-acid substitution model [31]. Node support was evaluated with 1000 ultrafast bootstrap replicates and 1000 SH-aLRT tests.

### Viral metagenome diversity and abundance in Triatominae

Diversity analyses were conducted with an abundance-based matrix derived from 23 tissue samples, built using read counts assigned to identified viral non-redundant contigs, hereafter referred to as viral Operational Taxonomic Units (vOTUs). The read counts were obtained by mapping quality-filtered reads to the vOTUs collection through the BBmap tool (v39.01). To avoid false positives, we used a minimum threshold of 99% nucleotide identity for read mapping and >75% read coverage across the entire length of each vOTU [32]. Subsequently, the abundance matrix was normalized based on sequence length and library size (quality-filtered reads per sample). This normalization was estimated as Reads Per Kilobase per Million mapped reads (RPKM).

The diversity analyses considered four factors: 1) species (*T. rubida*, *T. sanguisuga, T. indictiva*, *T. gersteackeri*, and *H. protracta*), 2) tissue type (GUT, GONADS and ABD), 3) geographical location (Arizona, New Mexico, and Texas), and 4) sex (male and female). Alpha diversity was evaluated using the Richness (S), Shannon-Weaver (H), Simpson (1-D), and Evenness (E) indices, while Beta diversity was calculated using Bray-Curtis (BC) and Euclidean distances. Statistical differences between defined groups were determined through analysis of similarities (ANOSIM) and permutational multivariate analysis of variance (PERMANOVA). To identify the vOTUs contributing the most to the dissimilarity between groups, similarity percentages analysis (SIMPER) was employed. All diversity indices and statistical tests were calculated using R [33] with the metagenome-Seq [34], Vegan [35], and ggplot2 [36] packages, as well as PAST software (version 4) [37]. To visualise overlap among viral OTUs detected in the studied Triatominae, we first transformed the vOTU list (S3 Table) into a binary presence-absence matrix with a custom-made R script. The binary matrix was passed to the UpSetR package (v1.4.0) [38] to create an UpSet plot showing the unique and shared vOTUs across Triatominae species.

## Results

Sequencing of 23 Triatominae tissue samples generated a total read length of 277 Gb. Following quality control, the reads were *de novo* assembled into 6,748,633 contigs. Viral RNA constituted a mean of 0.307% of the total RNA per sample (post rRNA removal), with an interquartile range (IQR) of 0.019%-0.848%. Detailed sequencing and assembly metrics for each tissue sample are presented in (S4 Table). For ten Triatominae individuals, dominant blood meal analysis identified five different hosts, including wood rat, opossum, toads (likely *Incilius nebulifer*), and human (S2 Table). While we cannot reliably exclude the possibility of human DNA contamination during the collection and dissection process, the results unequivocally show that all analyzed individuals of *T. rubida* and *H. protracta* fed on *Neotoma albigula*, whereas some *T. indictiva* and *T. gerstaeckeri* utilized toads and opossums as blood meal sources.

### Triatominae virome composition and virus sequences annotation

The *de novo* assembly revealed 41 unique vOTUs, which included 40 complete or partial genomes of novel viruses. Most of the Triatominae virome consists of complete or partially complete genomes of single-stranded RNA (ssRNA) viruses. Among these, six negative-sense ssRNA can be reliably assigned to the families *Rhabdoviridae*, *Chuviridae*, *Arenaviridae*, *Xinmoviridae,* and *Orthomyxoviridae*, and the order *Elliovirales* (Fig 1). The positive-sense ssRNA viruses detected belonged to the families *Virgaviridae*, *Benyviridae*, *Tombusviridae*, *Solemoviridae, Astroviridae*, and *Narnaviridae*.

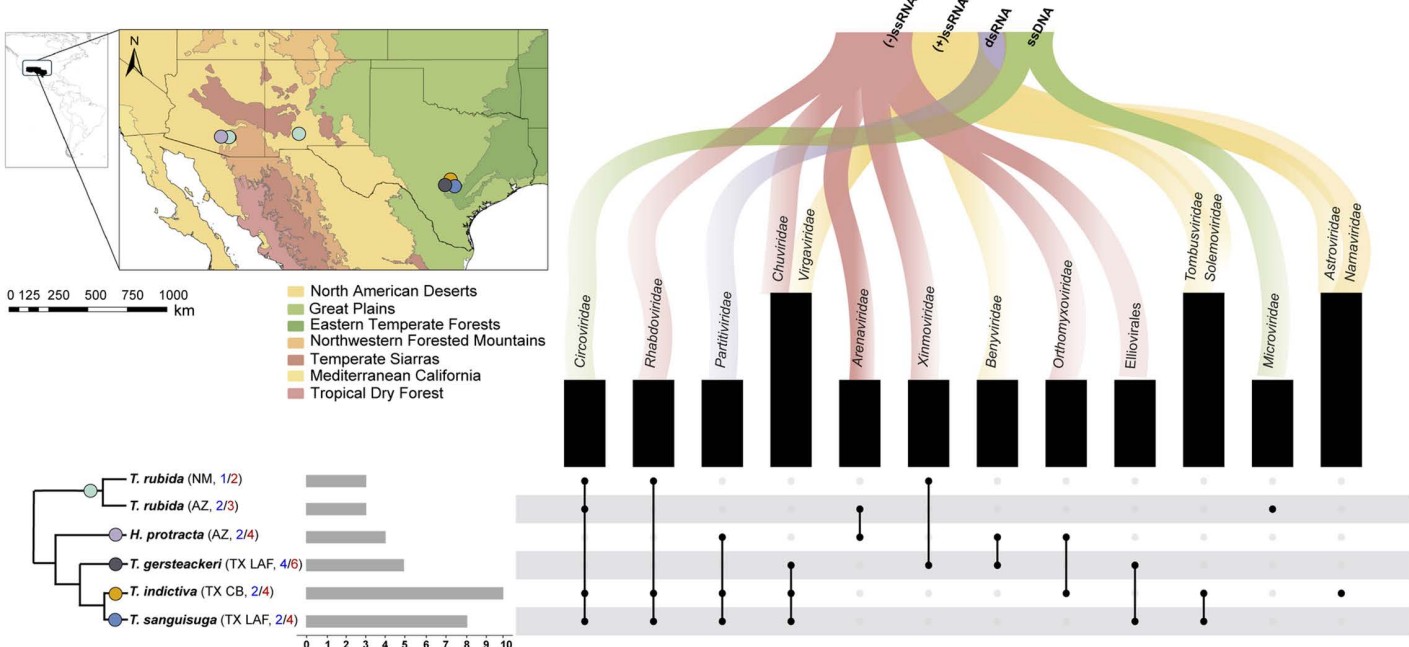

**Fig 1. Overview of the viral metagenome diversity associated with five Triatominae species sampled from Arizona (AZ), New Mexico (NM) and Texas (TX).** The schematic phylogeny represents Maximum Likelihood tree derived from eight mitochondrial genes with collapsed species/population clusters (S1 Fig). The abbreviations in parentheses following TX refer to two nearby sampling localities in Texas (detailed in S1 Table). The numbers analyzed per branch designate individuals in blue and tissue samples in red. The scale bar indicates the number of identified viral families and orders per branch. Map base layers: 1) Ecoregions: https://www.epa.gov/eco-research/ecoregions-north-america U.S. 2) Administrative boundaries: https://www.geoboundaries.org.

Additionally, a single double-stranded RNA (dsRNA) virus from the family *Partitiviridae* and two single-stranded DNA (ssDNA) viruses from the families *Circoviridae* and *Microviridae* were detected. All five Triatominae species harbor a diverse virome. We identified three to ten viral families, with *T. indictiva* and *T. sanguisuga* harboring the highest diversity (ten and eight families, respectively). The viral diversity at the individual sample level includes a median of four vOTUs, with an interquartile range (IQR) of four and a maximum of eleven vOTUs.

### Negative-sense, single-stranded RNA metavirome

A total of 17 vOTUs associated with negative-sense single-stranded RNA (ssRNA-) viruses were identified across the Triatominae tissue samples (Fig 2 and S3 Table). Two vOTUs from the *Arenaviridae* (Fig 2A) family detected in *H. protracta* and *T. rubida* clustered together with Big Brushy Tank Virus (BBTV) (ACL36816.1) in the phylogeny. BBTV was previously identified in the white-throated woodrat, *Neotoma albigula*, from Arizona, suggesting ecological link between these viruses, reservoir hosts and triatomine vectors in this region. Seven vOTUs were detected within the *Chuviridae* (Fig 2B) family. One vOTU from *T. sanguisuga* comprised a complete chuvirus genome and clustered as a sister taxon to Guiyang Chuvirus 1 (UHK03046.1), previously described from the paddy ear head bug (*Cletus punctiger*) in China. Six distinct vOTUs were identified in *T. indictiva, T. sanguisuga,* and *T. gerstaeckeri*. Phylogenetic comparison showed that these vOTUs are most closely related to Sanya Chuvirus 2 (UHK03098.1), detected in a brown planthopper *Nilaparvata lugens* in China. In addition, we identified two vOTUs, corresponding to the full-length RdRp segments of an Elliovirales (Fig 2C) genome. Both vOTUs found in *T. sanguisuga* and *T. gersteackeri* shared the highest similarity with the Orius laevigatus bunyavirus 2 (PP908622.1; S3 Table). The phylogenetic analysis places them as sister taxa to a clade

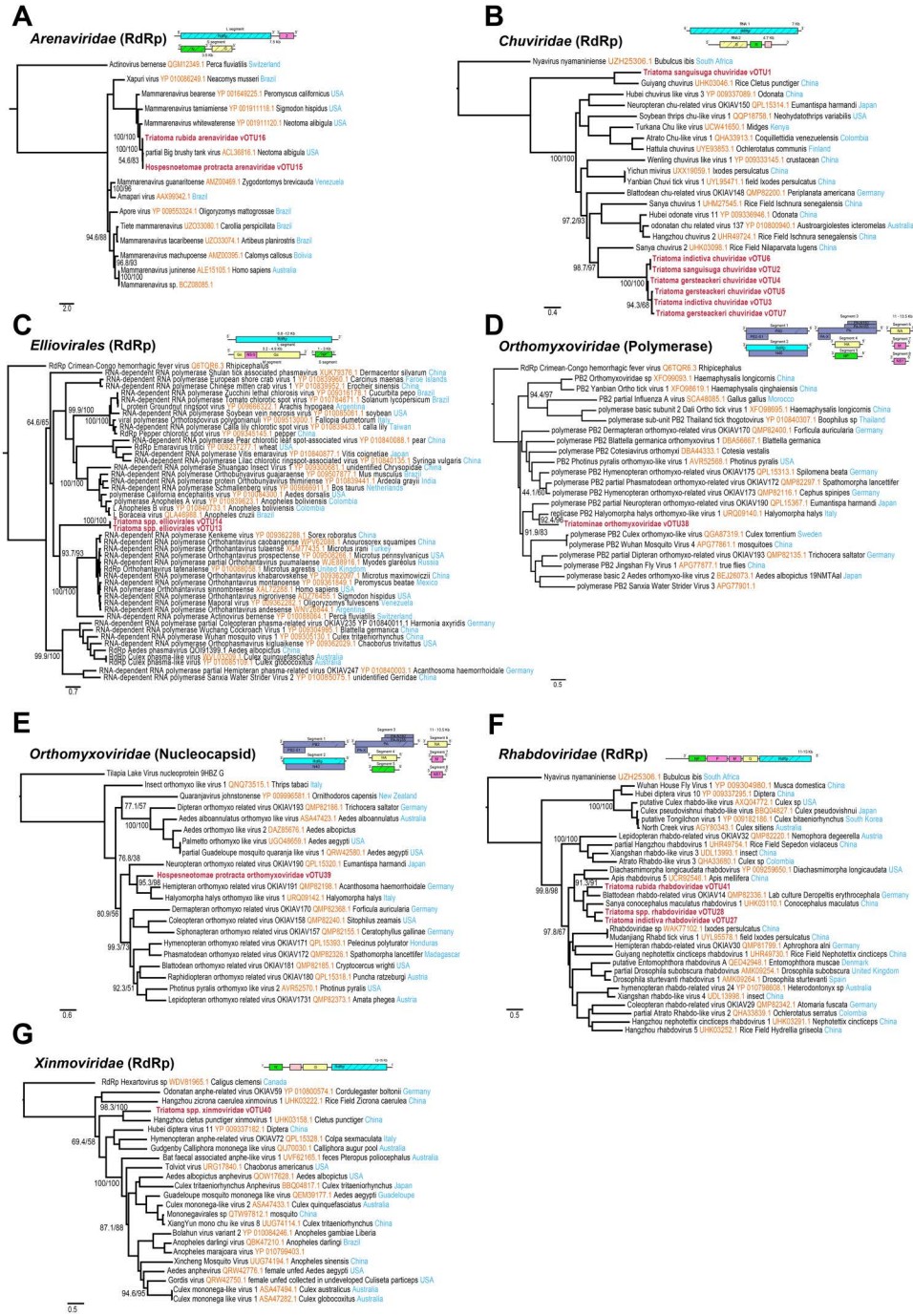

**Fig 2. Phylogenetic trees of negative-sense, single-stranded RNA viruses detected in Triatominae.** Maximum-likelihood trees were inferred from RdRp, Polymerase, and Nucleocapsid protein sequences of *Xinmoviridae, Arenaviridae, Chuviridae, Rhabdoviridae, Elliovirales,* and *Orthomyxoviridae.* Newly identified virus sequences in bold and support values evaluated with 1,000 ultrafast bootstrap replicates and 1,000 SH-aLRT tests are shown under nodes. Schematic diagrams depict the genomic architecture of representative viruses, with the hatched regions indicating the part of virome obtained in this study.

of Orthohantaviruses isolated from various rodents. Within the *Orthomyxoviridae* (Fig 2D–2E) family, we assembled two vOTUs corresponding to partial polymerase and nucleocapsid segments. The first, detected in both *H. protracta* and *T. indictiva* samples, with the closest relative Halyomorpha halys orthomyxo-like virus 1 (URQ09140.1). The phylogenetic analysis placed the virus within hemipteran and neuropteran orthomyxoviruses. Similar phylogenetic placement was recovered for the second vOTU found exclusively in *H. protracta*, which showed the highest similarity to Hemipteran Orthomyxo-related Virus OKIAV191 (QMP82198.1), previously isolated from hawthorn shield bug *Acanthosoma haemorrhoidale* in Germany.

In the *Rhabdoviridae* (Fig 2F) family, three vOTUs were assembled from *T. indictiva, T. sanguisuga,* and *T. rubida*. Two vOTUs corresponded to RdRp-encoding segments closely related to Sanya Conocephalus Maculatus Rhabdovirus 1 (UHK03110.1), originally reported from bush crickets in rice fields in China, and one vOTU to Blattodean Rhabdo-related Virus OKIAV14 (QMP82336.1), previously detected in red-headed cockroaches from Germany. Finally, in the *Xinmoviridae* (Fig 2G) family*,* partial RdRp segments were assembled from *T. gerstaeckeri* and *T. rubida* samples showing similarity to Hangzhou Cletus punctiger Xinmovirus 1 (UHK03158.1), a virus originally characterized from rice stink bug *Cletus punctiger* in China.

## Positive-sense, single-stranded RNA metavirome

A total of 15 vOTUs associated with positive-sense single-stranded RNA (ssRNA+) viruses were identified (Fig 3 and S3 Table). A single vOTU associated with the partial genome of the family *Astroviridae* (Fig 3A) was assembled from *T. indictiva*. This vOTU showed similarity to Flumine Astrovirus 3 (UQB76172.1), which was sampled from a natural water source in New Zealand. However, although the phylogenetic analysis placed the vOTU within the "Flumine astrovirus cluster", its position within the tree remained unclear due to low support (Fig 3A).

Four vOTUs were identified in the family *Benyviridae* (Fig 3B), with two from *T. gerstaeckeri* and two from *H. protracta*, all clustering with Sanya Benyvirus 1 (UHK03121.1), previously isolated from *Sesamia inferens* in China. We assembled two complete *Narnaviridae* (Fig 3C) genomes, from *T. indictiva*. One vOTU showed sequence identity and clustered with Serbia Narna-like Virus 3 (QNS17454.1), found in *Culex pipiens* from Serbia. The other full segment showed nucleotide similarity to Wuhan spider virus 7 (NC_033702.1), however the phylogenetic analysis placed this vOTU with Kummerowia striata ourmiavirus (QKI29250.1) with low support values. Two unrelated vOTUs corresponding to complete genomes from the family *Solemoviridae* (Fig 3D) were also identified. One vOTU, from *T. indictiva*, contained a fully assembled RdRp sequence showing high sequence identity Atrato Sobemo-like Virus 2 (QHA33888.1), previously identified in *Culex* sp. from Colombia. The second vOTU is related to both Rhodnius prolixus Virus 5 and 6 (QYC92635.1, QYC92637.1), found previously in ovaries of *Rhodnius prolixus* from Brazil.

Additionally, in both *T. sanguisuga* and *T. indictiva*, a unique vOTU related to the family *Tombusviridae* (Fig 3E) was identified. This vOTU corresponded to a complete genome that showed similarity to Hemipteran Tombus-related Virus (QTJ63611.1) from elongated-grass bugs, however its position within the topology was not well supported. Finally, five vOTUs were identified as *Virgaviridae* (Fig 3F), which include both complete and partial genomes. Three vOTUs recovered from *T. gersteackeri* exhibited similarity to Xiangshan Martelli-like Virus 3 (UDL14011.1) and Xiangshan martelli-like virus 2 (UDL14009.1), previously identified from an insect collected in China. The remaining two vOTUs were recovered from *T. sanguisuga* and *T. indictiva and* showed similarity to the Pedersore Virga-like Virus (UYL94368.1) and Ginka Virga-like Virus (UYE93845.1), which were identified in *Ochlerotatus communis* in Sweden and Finland. Phylogenetically, all these vOTUs create a monophyletic cluster closely related to the viruses mentioned above (Fig 3F).

## Single-stranded DNA metavirome

Five vOTUs were identified, belonging to two families of ssDNA viruses, *Circoviridae* and *Microviridae* (Fig 4 and S3 Table). Four vOTUs were assigned to the *Circoviridae* (Fig 4A–4B) family. Three of these vOTUs detected in *T. rubida*,

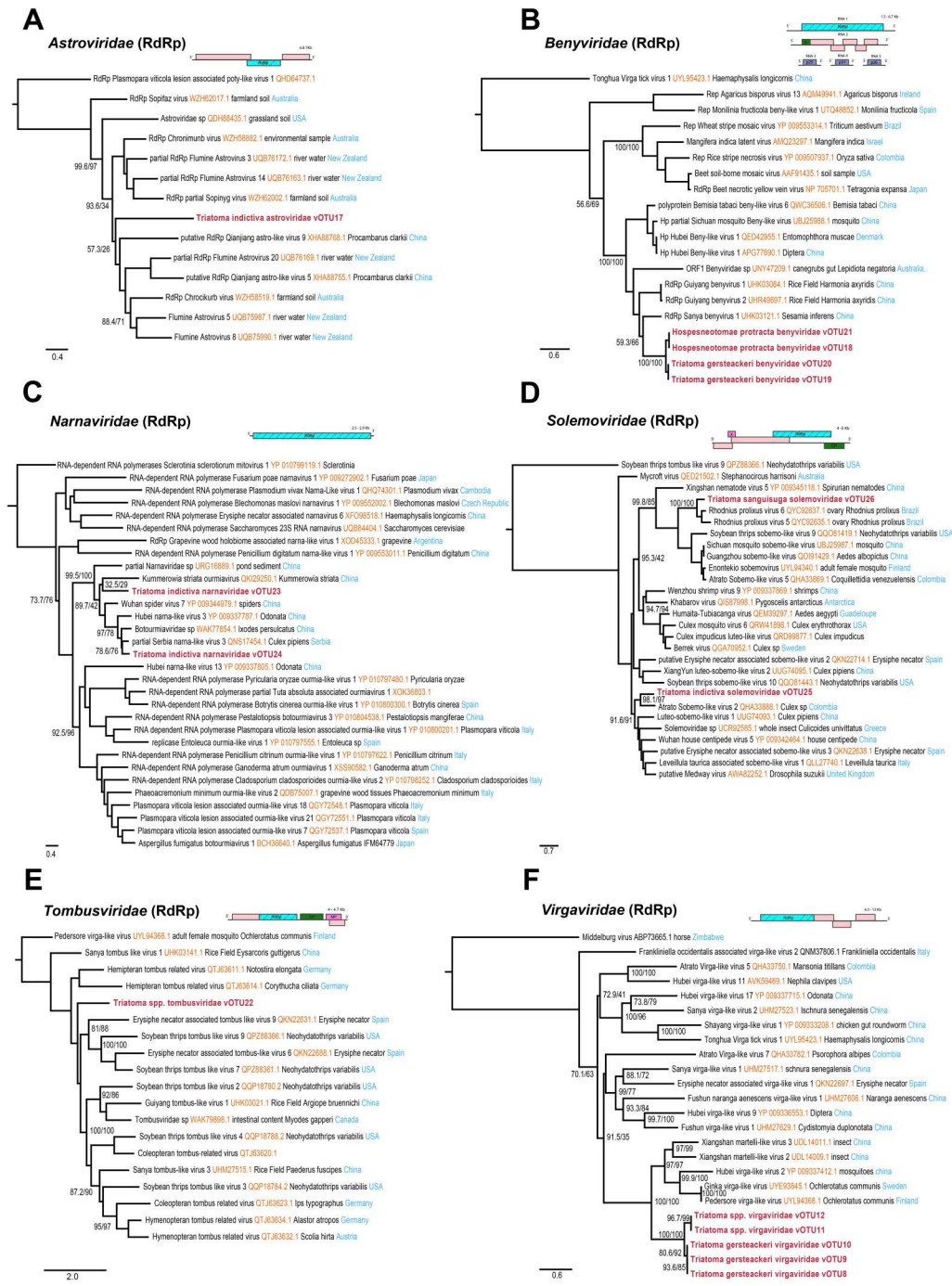

**Fig 3. Phylogenetic trees of positive-sense, single-stranded RNA viruses found in Triatominae.** Maximum-likelihood trees were reconstructed from RdRp protein sequences of *Virgaviridae, Benyviridae, Solemoviridae, Astroviridae, Tombusviridae,* and *Narnaviridae.* Node support was evaluated with 1,000 ultrafast bootstrap replicates and 1,000 SH-aLRT tests, and viruses identified in this study are highlighted in bold. Schematic diagrams represent the genomic architecture of each virus family, with the hatched regions indicating the part of virome obtained in this study.

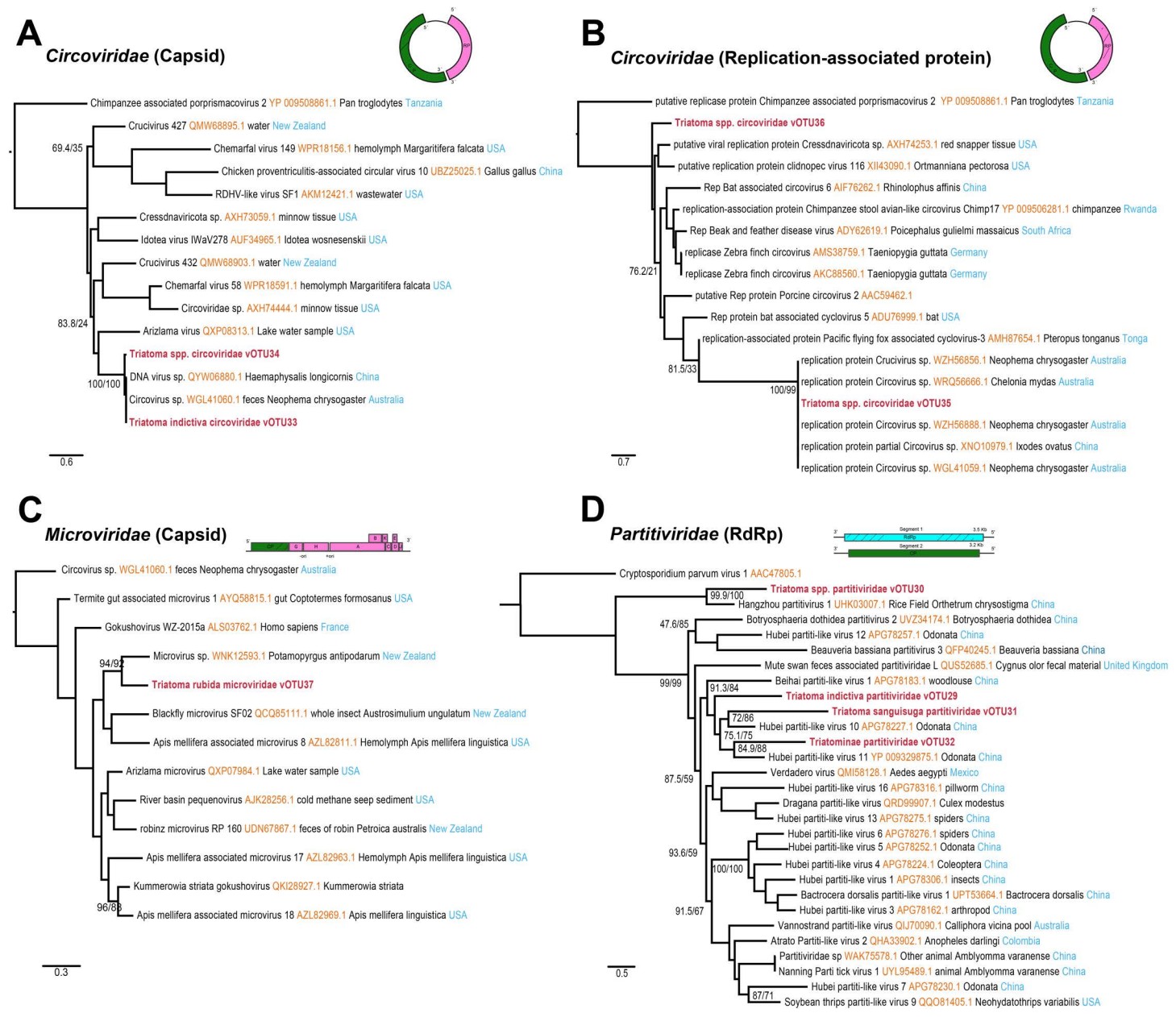

**Fig 4. Phylogenetic trees of single-stranded DNA and double-stranded RNA viruses found in Triatominae.** Maximum-likelihood trees were reconstructed from RdRp protein sequences of *Partitiviridae,* Capsid protein of *Microviridae* and *Circoviridae,* and replication-associated protein of *Circoviridae.* Identified viruses in this study are highlighted in bold letters. Node support was evaluated with 1,000 ultrafast bootstrap replicates and 1,000 SH-aLRT tests and is shown in branches. Schematic diagrams represent the genomic architecture of each virus family, with the shaded regions indicating the sequence areas successfully recovered in this study.

*T. sanguisuga,* and *T. indictiva*, showed high similarity to a capsid (WGL41060.1) and replicase segments (XNO10979.1) of Circovirus sp., which were sampled from the feces of *Neophema chrysogaster* in Australia and *Ixodes ovatus* from China. The additional vOTU was found in *T. rubida* and *T. indictiva*. It showed similarity to Cressdnaviricota sp. (AXH74253.1), however its phylogenetic position remained unstable (Fig 4B). The *Microviridae* (Fig 4C) family was

exclusively detected in *T. rubida* and was found to be related to Microvirus sp. (WNK12593.1), sampled from the New Zealand mud snail.

## Double-stranded RNA metavirome

Only one family of double-stranded RNA viruses, *Partitiviridae* (Fig 4D), was identified in the Triatominae samples. Partial genome segments were assembled into four distinct vOTUs from the species *T. sanguisuga*, *T. indictiva*, and *H. protracta* (Fig 4 and S3 Table). Three of these vOTUs, exhibited high similarity to Hubei Partiti-like Virus 10 (APG78227.1) and Hubei Partiti-like Virus 11 (YP_009329875.1) both the viruses were previously reported in dragonflies from China. The fourth vOTU, detected both in *T. sanguisuga,* and *T. indictiva,* was closely related to Hangzhou partitivirus 1 (UHK03007.1), reported in dragonflies found in rice fields in China.

## Geographic specificity of the Triatominae metavirome

Among the 41 vOTUs identified in this study, 26 were found exclusively in Texas, while five were unique to Arizona. In Arizona, we found vOTUs belonging exclusively to the *Arenaviridae* and *Microviridae* families detected in *T. rubida* and *H. protracta*. Whereas, in Texas, vOTUs exclusively belonged to the order Elliovirales and the families *Chuviridae*, *Tombusviridae*, *Virgaviridae*, *Astroviridae*, *Solemoviridae*, and Narna-related viruses. Notably, two vOTUs were identified in triatomines from three states, corresponding to ssDNA viruses of the *Circoviridae* family. Some vOTUs showed high relative abundance in various samples, particularly those related to *Solemoviridae*, and Elliovirales (S3 Table and S2 Fig).

Although 31 of the 41 vOTUs were geographically specific, alpha diversity indices, as measured by the Shannon diversity index, showed slight variation among specimens from the three states (mean Shannon index = 0.675, SD = 0.398), but no statistically significant differences were observed (Kruskal-Wallis test, $p > 0.05$). Similarly, richness and evenness metrics did not differ significantly between states (Kruskal-Wallis test, $p > 0.05$), indicating a broadly uniform distribution of alpha diversity indices across geographical locations. In contrast, beta diversity analyses revealed some geographic structuring. Bray-Curtis-based Non-metric multidimensional scaling (NMDS) and Euclidean-based heatmaps suggested distinct patterns in the composition and abundance of vOTUs by state. These observed groupings were statistically significant (PERMANOVA test, $p = 0.003$, $R^2 = 0.171$) (S2 Fig).

## Species-specific metavirome profile in triatomines

Considering that some Triatominae were collected exclusively in Texas and Arizona, and the observed geographic patterns might also reflect species effect in shaping the virome, we further investigated whether vOTUs shared across locations exhibited species or lineage-specific patterns. Out of the 41 vOTUs identified, 24 were detected exclusively in a single species: nine in *T. gerstaeckeri*, seven in *T. indictiva*, three each in *T. rubida* and *H. protracta*, and two in *T. sanguisuga*. The remaining 17 vOTUs were not species-specific and were shared among different species. None of these vOTUs were found in all five species. Specifically, eight vOTUs were shared between *T. indictiva* and *T. sanguisuga*; two of these vOTUs, belonging to the families *Circoviridae* and *Partitiviridae,* were shared with *T. rubida* and *H. protracta*, respectively. The other non-species-specific vOTUs were exclusively shared between pairs of species: *T. rubida* had a unique vOTU shared with *T. indictiva*, *T. sanguisuga,* and *T. gerstaeckeri*. Similarly, *T. indictiva* and *H. protracta* shared one vOTU, while *T. gerstaeckeri* and *T. sanguisuga* had two vOTUs in common (Fig 5).

The Shannon diversity index of vOTUs revealed no significant differences among Triatominae species (Kruskal-Wallis test, $p > 0.05$). In contrast, significant differences were observed in vOTUs richness and evenness across the five Triatominae species (Kruskal-Wallis test, $p < 0.05$). Beta diversity analyses revealed distinct clustering patterns in vOTUs composition and abundance among species, with statistically significant groupings (PERMANOVA test, $p = 0.001$, $R^2 = 0.502$; ANOSIM test, $p = 0.0001$, $R = 0.713$) (Fig 5). Notably, samples from different Triatominae species generally clustered distinctly, and the virome of *T. indictiva* exhibited the highest diversity of vOTUs and clustered more distant from the virome of other species, aligning more closely with *T. sanguisuga*, as these two species shared the most vOTUs.

**Fig 5. Alpha and beta diversity of vOTUs in Triatominae natural populations from southern United States. A)** NMDS based on Bray-Curtis distances, showing clustering patterns by Triatominae species (Stress = 0.097, PERMANOVA, $p < 0.001$; $R^2 = 0.502$). **B)** Shannon diversity index of vOTUs by Triatominae species (Kruskal-Wallis test, $p = 0.399$). **C)** Venn diagram displaying the number of shared vOTUs among Triatominae species. **D)** Heatmap based on normalized read counts of vOTUs and their Euclidean-based clustering among Triatominae species.

The SIMPER analysis showed that a group of six vOTUs explained more than 74% of the dissimilarity in the virome composition among the species, highlighting the contribution of the species-specific vOTUs. Solemo_vOTU25, unique to *T. indictiva*, was the main contributor, explaining 26.8% of the dissimilarity. It was followed by Solemo_vOTU26 (14.7%), found exclusively in *T. sanguisuga*. Additionally, Rhabdo_vOTU41 was present only in *T. rubida*, while Arena_vOTU15

was unique to *H. protracta*. Narna_vOTU24, contributing 6.3% to the dissimilarity, was also found only in *T. indictiva*, and Beny_vOTU19, with a contribution of 3%, was unique to *T. gerstaeckeri*. Furthermore, the Ellio_vOTU13 and Ellio_vOTU14 contributed an additional 11.37% and 7.89%, respectively, and were more abundant in *T. sanguisuga* and *T. gerstaeckeri* (S5 Table).

### Sex, tissue, and *T. cruzi* infection status determinants

A uniform distribution of alpha viral diversity indices was observed across tissue type, sex, and *T. cruzi* infection status. Shannon, richness, and evenness indices did not show statistically significant differences among these factors (Kruskal-Wallis tests, $p > 0.05$). L5 specimens exhibited a higher Shannon index (Mean Shannon index = 1.18, SD = 0.163) compared to males (Mean Shannon index = 0.548, SD = 0.408). The Beta diversity analyses revealed no distinct clustering patterns in vOTU composition or abundance based on sex, tissue type, or *T. cruzi* infection status (PERMANOVA test, $p > 0.05$; ANOSIM test, $p > 0.05$) (S2 Fig).

Remarkably, 24 of the vOTUs identified in this study were found in reproductive tissue. Additionally, two vOTUs from *T. cruzi*-positive samples, which belong to the viral families *Orthomyxoviridae* and *Chuviridae*, were not exclusive to the infected individuals.

## Discussion

Our study advances the field by characterizing the virome of five Triatominae species from natural populations in the southern United States: *Triatoma rubida, Hospesneotomae protracta, Triatoma indictiva, Triatoma sanguisuga*, and *Triatoma gerstaeckeri*. We identified a diverse range of viral metagenomes, including negative- and positive-sense ssRNA, dsRNA, and small circular ssDNA viruses from various families. Our findings suggest that viral composition and abundance are primarily shaped by Triatominae species identity, highlighting a species-specific viral metagenome profile.

A significant proportion of the identified sequences belonged to *Chuviridae, Virgaviridae, Benyviridae, Partitiviridae,* and *Rhabdoviridae.* While some members of these families infect non-arthropod hosts, the phylogenetic analyses showed that most viral sequences clustered within clades of previously described arthropod-associated viruses, reinforcing their specificity to insects. Additionally*,* these viral taxa have been consistently reported in insect viromes [2,39,40] and, in some cases, as endogenous viral elements [41–44], further supporting their specificity to arthropods.

Interestingly, viral sequences from families primarily associated with plant infections were also detected. Solemoviruses, in particular, have been repeatedly identified in triatomines. Rhodnius prolixus virus 5 (RpV5) and Rhodnius prolixus virus 6 (RpV6), initially reported in *R. prolixus* [14], have since been found in *Rhodnius robustus* in French Guiana and *Rhodnius neglectus* in Brazil [15]. This study expands the known host range of *Solemoviridae* by identifying Triatoma indictiva solemoviridae vOTU25 and Triatoma sanguisuga solemoviridae vOTU26. The recurring presence of these viruses in multiple triatomine species across different regions suggests a widespread distribution throughout the Americas. Additionally, we identified several plant-virus-related sequences, including Triatoma spp. tombusviridae vOTU22 and Benyviruses (Beny_vOTU18-21) in *T. gerstaeckeri* and *H. protracta*. While *Benyviridae* members are primarily associated with plant infections [45], their detection in insects has been documented [46–48], though their role in non-plant hosts remains unclear [49]. The presence of plant-associated viruses in Triatominae metaviromes may appear unexpected, however, it is not implausible. Ability of triatomines to ingest plant-derived sugars have been previously recorded, and they retain functional α-amylase genes necessary for starch digestion [50]. Although our study does not evaluate whether triatomines can act as vectors of plant viruses, this possibility cannot be excluded, particularly given that host accessibility strongly shapes kissing-bug feeding patterns and previous reports indicate occasional plant feeding in Triatominae [50,51].

Vertebrate-associated viruses within the family *Arenaviridae* were also identified, specifically Hospesneotomae protracta arenaviridae vOTU15 and Triatoma rubida arenaviridae vOTU16. These viruses are phylogenetically related to

BBTV, previously isolated from *Neotoma albigula* in Arizona [52,53]. BBTV belongs to the genus *Mammarenavirus*, which includes 51 species primarily associated with rodents [54]. Notably, these arenaviruses were detected exclusively in *T. rubida* and *H. protracta* from Arizona. Consistent with these findings, we detected *Neotoma* blood in all analyzed individuals of *T. rubida* and *H. protracta*, further supporting the association between arenavirus sequences and rodent-derived blood meals. The presence of *Arenaviridae* vOTUs in the intestinal tract suggests that triatomines may act as mechanical carriers, dispersing viral particles through feces when interacting with rodent hosts. However, it remains unclear whether triatomines contribute to the natural transmission cycles of *Mammarenavirus*, or whether these findings simply reflect transient viral passage. Further investigation is needed to elucidate the role of triatomines in *Mammarenavirus* epidemiology.

In addition to RNA virome, we identified small circular DNA viruses from *Circoviridae* and *Microviridae* family. Circovirus sequences were phylogenetically related to viruses previously found in environmental samples and insects. Although most known *Circoviridae* members infect vertebrates [55] *Cyclovirus* species have also been identified in arthropods [56]. Their presence in hematophagous insects may be linked to the ingestion of viremic blood [57]. Additionally, circoviruses have been detected circulating in vertebrates from the study region [58,59], which could explain their presence in triatomines. In contrast, *Microviridae* are bacteriophages, likely representing components of the bacterial microbiome within triatomines. These bacteriophages have been consistently detected in insect microbiomes [60,61], including species from regions examined in this study [62].

Analysis of viral diversity and abundance revealed significant differences in vOTU richness and evenness among the five Triatominae species. Beta diversity analyses consistently showed species-specific clustering patterns, while sex and tissue type had minimal influence. These findings align with previous studies on mosquitoes and ticks, where virome richness and diversity primarily vary at the species level [3,63–68]. Beyond species identity, host interactions may also play a role in shaping virome composition in Triatominae, as observed in other hematophagous arthropods such as mosquitoes and ticks, where host associations significantly influence virome structure [3,66]. Here, the detection of viral sequences from the families *Solemoviridae* and *Arenaviridae* suggests that host interactions contribute to virome composition in Triatominae. Notably, Triatoma indictiva solemoviridae vOTU25 and Triatoma sanguisuga solemoviridae vOTU26 were identified as key drivers of differences in viral abundance and composition among the Triatominae species.

Another factor known to influence insect virome composition is geography [5,66,69]. Our analysis revealed that although 31 of the 41 vOTUs were geographically specific, beta diversity analyses identified significant geographic structuring among specimens from Arizona, Texas, and New Mexico. Nevertheless, since some Triatominae species were collected exclusively in Texas and Arizona, the observed geographic patterns may, in part, reflect the influence of Triatominae species identity on virome composition. These findings suggest that while overall viral richness and evenness remain relatively stable across regions, differences in virome composition may primarily be shaped by species identity, as mentioned earlier, without ruling out the complex interplay of localized environmental factors, host distributions, and ecological interactions. Future research should aim to disentangle the relative contributions of these factors, refining our understanding of virus-host dynamics in Triatominae.

Finally, a key challenge in viral metagenome studies is differentiating actively replicating viruses from endogenous viral elements (EVEs) [70]. EVEs arise from viral insertions into host genomes, complicating their distinction from exogenous viruses [71]. Although complete RNA virus genomes can occasionally integrate into a host genome, such events are relatively rare. EVEs are typically fragmented viral sequences with disrupted ORFs [72]. In our study, however, most of the viral sequences we identified are full-length or nearly full-length coding regions of RNA viral genomes. This suggests that these sequences are unlikely to represent EVEs expressed by the host. In addition to challenges related to EVEs, our study is constrained by the relatively small sample size, which may limit the generalizability of our findings, particularly regarding the effects of geographic variation on viral abundance and composition in triatomine natural populations. Overcoming these limitations will require expanded sampling efforts together with complementary approaches, such as whole-genome and small RNA sequencing, to assess viral replication and host antiviral responses [65,73]. Integrating

these methodologies in future work will improve the distinction between EVEs and actively replicating viruses in Triatominae and provide deeper insight into virus–host interactions. Moreover, broader sampling across the geographic and ecological ranges of triatomine species will be essential for any robust conclusions on the factors that shape viral diversity in these important disease vectors.

## Conclusion

This study offers the most comprehensive characterization of Triatominae viromes to date in the United States, significantly advancing our understanding of viral diversity, species specificity, and potential ecological interactions in these vectors. We reveal species-specific virome patterns and show that Triatominae identity is a dominant factor shaping viral composition. The detection of viruses associated with insects, plants, and vertebrates highlights the intricate nature of viral interactions within these hematophagous hosts. Given this complexity, together with the limited number of species and individuals analyzed so far, further research will be essential to explore the functional implications of these viral components and their potential impact on Triatominae ecology and vector competence.

## Supporting information

**S1 Table. Sample metadata and *T. cruzi* meta-assembly screening.** Summary of 23 Triatominae individuals with species, sex, geographic location, tissue type, QC, and *T. cruzi* detection.
(PDF)

**S2 Table. Blood meal analysis of Triatominae gut tissues.** Twelve DNA isolates from gut samples analysed using vertebrate 12S rRNA gene sequences to identify blood meal sources.
(PDF)

**S3 Table. Viral operational taxonomic units (vOTUs).** List of vOTUs detected in Triatominae natural populations from the southern United States.
(PDF)

**S4 Table. Sequencing metrics.** Summary of raw sequencing reads, reads retained after quality control, total number of assembled contigs, and number of viral reads per sample.
(PDF)

**S5 Table. SIMPER analysis.** Contribution of each Viral Operational Taxonomic Units (vOTUs) to the Bray-Curtis dissimilarities observed in Triatominae natural populations between species (*H. protracta, T. rubida, T. indictiva, T. sanguisuga, T. gersteackeri*).
(PDF)

**S1 Fig. Phylogeny of sampled Triatominae.** Maximum-likelihood tree based on eight mitochondrial genes retrieved from each tissue sample analyzed in this study which correspond to five species (*Triatoma rubida, Hospesneotomae protracta, Triatoma gerstaeckeri, Triatoma sanguisuga,* and *Triatoma indictiva*). Multiple libraries from the same individual are shown as an internal quality-control measure to confirm consistent mitochondrial signal and detect potential issues such as sample mislabeling or multiplexing errors. Node labels indicate bootstrap support values (100 replicates).
(PDF)

**S2 Fig. Alpha and beta diversity of vOTUs in natural populations of *Triatoma*.** A) Non-metric multidimensional scaling (NMDS) based on Bray-Curtis distances, showing clustering patterns by study location (PERMANOVA, p = 0.003; $R^2$ = 0.171), sex (PERMANOVA, p = 0.426; $R^2$ = 0.093), and tissue (PERMANOVA, p = 0.97; $R^2$ = 0.138). B) Shannon diversity index of vOTUs by location (Kruskal-Wallis test, p = 0.816), sex (Kruskal-Wallis test, p = 0.320), and tissue

(Kruskal-Wallis test, p = 0.744). C) Venn diagrams displaying the number of shared vOTUs among triatomine populations, grouped by location, sex, and tissue type.
(PDF)

## Acknowledgments

We would like to acknowledge our collaborators that contributed to the sampling efforts Robert L. Smith, Walter Roachell and Alvaro Romero.

## Author contributions

**Conceptualization:** Eva Novakova.

**Formal analysis:** Insan Habib, Juan Camilo Hernandez-Valencia, Jana Martinu.

**Funding acquisition:** Eva Novakova.

**Methodology:** Juan Camilo Hernandez-Valencia, Jana Martinu, Eva Novakova.

**Project administration:** Eva Novakova.

**Supervision:** Eva Novakova.

**Validation:** Juan Camilo Hernandez-Valencia.

**Visualization:** Insan Habib.

**Writing – original draft:** Insan Habib, Juan Camilo Hernandez-Valencia, Eva Novakova.

**Writing – review & editing:** Insan Habib, Juan Camilo Hernandez-Valencia, Jana Martinu, Eva Novakova.

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
