## [Decision Letter · Decision Letter 0]

17 Nov 2025

Viral metagenome characterization reveals species-specific virome profiles in Triatominae populations from the southern United States

Dear Dr. Novakova,

Thank you for submitting your manuscript to PLOS Neglected Tropical Diseases. After careful consideration, we feel that it has merit but does not fully meet PLOS Neglected Tropical Diseases's publication criteria as it currently stands. Therefore, we invite you to submit a revised version of the manuscript that addresses the points raised during the review process.

Please submit your revised manuscript within by Jan 16 2026 11:59PM. If you will need more time than this to complete your revisions, please reply to this message or contact the journal office at plosntds@plos.org. Please include the following items when submitting your revised manuscript:

We look forward to receiving your revised manuscript.

Kind regards,

Adly M.M. Abd-Alla, Prof asso.

Section Editor

Adly Abd-Alla

Section Editor

Shaden Kamhawi

co-Editor-in-Chief

Paul Brindley

co-Editor-in-Chief

**Journal Requirements:**

1) Please provide an Author Summary. This should appear in your manuscript between the Abstract (if applicable) and the Introduction, and should be 150-200 words long. The aim should be to make your findings accessible to a wide audience that includes both scientists and non-scientists. Sample summaries can be found on our website under Submission Guidelines:

2) Some material included in your submission may be copyrighted. According to PLOSu2019s copyright policy, authors who use figures or other material (e.g., graphics, clipart, maps) from another author or copyright holder must demonstrate or obtain permission to publish this material under the Creative Commons Attribution 4.0 International (CC BY 4.0) License used by PLOS journals. Please closely review the details of PLOSu2019s copyright requirements here: PLOS Licenses and Copyright. If you need to request permissions from a copyright holder, you may use PLOS's Copyright Content Permission form.

Potential Copyright Issues:

- Figure 1. Please (a) provide a direct link to the base layer of the map (i.e., the country or region border shape) and ensure this is also included in the figure legend; and (b) provide a link to the terms of use / license information for the base layer image or shapefile. We cannot publish proprietary or copyrighted maps (e.g. Google Maps, Mapquest) and the terms of use for your map base layer must be compatible with our CC BY 4.0 license.

3) Please amend your detailed Financial Disclosure statement. This is published with the article. It must therefore be completed in full sentences and contain the exact wording you wish to be published.

**Reviewers' Comments:**

Reviewer's Responses to Questions

**Key Review Criteria Required for Acceptance?**

**Methods**

-Are the objectives of the study clearly articulated with a clear testable hypothesis stated?

-Is the study design appropriate to address the stated objectives?

-Is the population clearly described and appropriate for the hypothesis being tested?

-Is the sample size sufficient to ensure adequate power to address the hypothesis being tested?

-Were correct statistical analysis used to support conclusions?

-Are there concerns about ethical or regulatory requirements being met?

Reviewer #1: The objectives of the study are clearly stated, and the study addresses an important knowledge gap in Triatominae virome characterization. However, the sampling design requires clarification. The manuscript sometimes uses “sample” to denote sequencing libraries and other times to denote individuals (insects), which complicates interpretation. Inconsistent suffix labeling (e.g., “gonads,” “ovary,” “Agonad,” “BOVA,” “testes,” “abdomen,” “BGUT,” “gut,” “ovaries”) of tissues across figures and supplementary materials further obscures whether multiple libraries from the same insect were treated as independent observations. If so, this compromises assumptions of independence in statistical tests and should either be avoided or acknowledged. The number of insects included is mentioned, but the treatment of multiple tissue types per insect in diversity analyses is not explicit. This ambiguity weakens confidence in the robustness of the statistical framework. I recommend the authors clarify definitions of “sample” and “triatomine individual,” specify the analytical unit used for diversity analyses, and consider alternative approaches (e.g., aggregating libraries per individual or treating tissue as a dependent variable).

Considering the large number of variables included (triatomine species, area, predominant host species, stage/sex, tissue type, and T. cruzi infection status) in relation to the small sample size (23 libraries derived from tissues of approximately 11–16 triatomine individuals), the statistical power is limited. Blood meal identification was performed using Sanger sequencing of 12S fragments, which typically detects only the dominant host. Notably, overlapping host signals observed in Supplementary Table S2 suggest the presence of mixed templates, and recent metabarcoding studies have shown that multiple blood meals are frequent in triatomines. This methodological limitation should be acknowledged in the discussion. As illustrated in Figure 5, the clustering patterns and vOTU distributions should therefore be interpreted with caution, reinforcing that these results are exploratory rather than conclusive.

It is not entirely clear how rRNA removal was performed prior to downstream analyses. While the results are reported as “after removal of rRNA” (Ln. 195), the methods section does not specify whether this step was carried out experimentally (e.g., using an rRNA depletion kit during library preparation) or computationally (bioinformatic filtering of rRNA reads).

At Ln. 455–465, the authors report that beta diversity analyses revealed species-specific clustering patterns, while sex and tissue type had minimal influence. However, it is not clear what the unit of analysis was for beta diversity — whether the analyses were conducted at the level of the 23 nucleic acid libraries, aggregated by individual insect, or directly by species. Explicitly stating the analytical unit would improve transparency and allow readers to better evaluate the robustness of the diversity patterns reported. No major ethical or regulatory concerns were identified.

Reviewer #2: The hypothesis and objectives of the study are clearly stated and articulated, also well developed step by step in the methodology section. The population is clearly described, altought the number of samples is small. The samples info is detailed in Table S1, but it would be appreciated if you could add a few words in Sample set section where you describe how many specimens you collected per each sampling state, and also explain why you selected those 2 females for whole abdomen. The statistical analysis is fine for a good approximation to test the hypothesis.

**Results**

-Does the analysis presented match the analysis plan?

-Are the results clearly and completely presented?

-Are the figures (Tables, Images) of sufficient quality for clarity?

Reviewer #1: The results are presented in detail, but some issues require clarification:

References to figures are occasionally confusing. For example, Ln. 393–400 describes viral diversity but cites Supplementary Figure S1, which actually shows mitochondrial triatomine phylogeny. This must be corrected.

In Supplementary Figure S1, multiple libraries from the same insect appear as independent COI terminals, which does not provide additional phylogenetic information and risks confusing readers. Either one representative per insect should be shown, or the rationale should be justified.

In Supplementary Figure S2, beta diversity analyses are reported by location, but species identity and location are not independent. For instance, T. rubida was collected in both Arizona and New Mexico, while other species were restricted to single sites. This collinearity complicates interpretation, and the authors should clarify whether clustering reflects species, geography, or both.

Supplementary Table S4 shows large variance in viral read proportions (<0.001% to >5%), which deserves explicit discussion as it may reflect both biology and technical bias.

Overall, figures and tables are of sufficient quality but would benefit from more standardized labeling (e.g., tissues named consistently as “gut,” “gonads,” “ovaries,” etc.), as previously said.

Reviewer #2: The section present the results clearly and it was a wise decision to divide the results at first according to the viruses category, it makes the reading easier.

The figures are clear, but (after download) they loose resolution and become pixelated when zoomed in. If possible, please consider providing higher-resolution versions to ensure readability.

The tables are complete, detailed and the information is well organized.

Figure S2 presents important data that supports part of the objectives of the study (section "Geographic specificity of the Triatominae metavirome"). Therefore, I recommend including Figure S2 in the main manuscript, considering it represents graphically the indices used to test the hypothesis, and they are well arranged to enhance the understanding of these results.

In line 393, T. cruzi is not in italics.

Regarding the discussion section:

Line 409 and 410: I recommend a little change: "(...) viral composition and abundance COULD BE/MAY BE primarily shaped (...)", as you described later in the discussion (line 471-473) that your findings SUGGEST (...) and (...) MAY primarily be shaped by species identity". This just to avoid overgeneralizations, considering that the dataset included a limited number of species and few specimens per species.

Also, please check the scientific names in italics since line 418, and in the whole document just in case.

Line 429-431  to this statement: "While our study does not assess whether triatomines contribute to plant virus transmission, our findings highlight a potential ecological role warranting further investigation", maybe you can complement with information regarding herbivory in Triatomines and the ecological implications, it could contribute to improve this part of your discussion and deepen a little further to help explain your findings, for example please consider these papers : https://doi.org/10.1186/s13071-016-1401-0 ; https://doi.org/10.1101/2023.03.03.530934 ; https://doi.org/10.1590/s0074-02762011000400016

**Conclusions**

-Are the conclusions supported by the data presented?

-Are the limitations of analysis clearly described?

-Do the authors discuss how these data can be helpful to advance our understanding of the topic under study?

-Is public health relevance addressed?

Reviewer #1: The conclusions are generally consistent with the data but should be presented more cautiously. With 23 libraries derived from ~11–16 insects, statistical power is limited, and treating multiple tissue libraries as independent units may bias results. The discussion should explicitly acknowledge these limitations. The findings are relevant to understanding viruses diversity in hematophagous vectors, but the impact would be strengthened by clarifying whether species identity or geography was the main driver of beta diversity.

Reviewer #2: I suggest: "(...)the most comprehesive characterization of Triatominae virome to date IN THE US".

Please state some limitations in the conclusion as well.

The conclusion is clear and detail the main findings that make the study relevant for public health.

**Editorial and Data Presentation Modifications?**

Reviewer #1: -Standardize terminology for tissues across figures, tables, and supplementary material.

-Revise figure legends to avoid misleading terminology (e.g., “Host phylogeny of sampled Triatominae species.” > “phylogenetic reconstruction of Triatominae species”). The term host may be misleading, as it is generally used to refer to vertebrate blood meal sources rather than insect vectors.

-Ensure consistent use of “sample” (=libraries?) in relation to “triatomine bugs”

Reviewer #2: Check the scientific names, some are not in italics, and, if possible, enhance the figures quality

**Summary and General Comments**

Reviewer #1: This is an original and valuable contribution, representing a considerable investment of effort and resources. The authors assembled viral contigs from Triatominae nucleic acid libraries, recovering a broad diversity of ssRNA(+), ssRNA(–), dsRNA, and ssDNA viruses and exploring their evolutionary relationships. This is novel and relevant for PLOS NTD. However, several issues limit interpretability: (i) lack of clarity in distinguishing “samples” from “triatomine individuals,” (ii) inconsistent nomenclature of tissue libraries, (iii) unclear statistical treatment of multiple libraries from the same insect, and (iv) potential collinearity between species and location in beta diversity analyses. Addressing these points would strengthen both methodological rigor and transparency. I therefore recommend Major Revision, with emphasis on clarifying sampling design, analytical units, and limitations of independence in statistical tests.

Reviewer #2: It is a well-designed an carefully executed study with meaningful findings for the field and public health. Although the sample size is small, it provides a solid and valuable basis for future research on this topic.

PLOS authors have the option to publish the peer review history of their article (what does this mean? ). If published, this will include your full peer review and any attached files.

**Do you want your identity to be public for this peer review?** For information about this choice, including consent withdrawal, please see our Privacy Policy .

Reviewer #1: No

Reviewer #2: No

**Figure resubmission:**
---

## [Decision Letter · Decision Letter 1]

13 Jan 2026

Dear Assoc. Prof. Novakova,

We are pleased to inform you that your manuscript 'Viral metagenome characterization reveals species-specific virome profiles in Triatominae populations from the southern United States' has been provisionally accepted for publication in PLOS Neglected Tropical Diseases.

Best regards,

Adly M.M. Abd-Alla, Prof asso.

Section Editor

Adly Abd-Alla

Section Editor

Shaden Kamhawi

co-Editor-in-Chief

Paul Brindley

co-Editor-in-Chief

Reviewer's Responses to Questions

**Key Review Criteria Required for Acceptance?**

**Methods**

-Are the objectives of the study clearly articulated with a clear testable hypothesis stated?

-Is the study design appropriate to address the stated objectives?

-Is the population clearly described and appropriate for the hypothesis being tested?

-Is the sample size sufficient to ensure adequate power to address the hypothesis being tested?

-Were correct statistical analysis used to support conclusions?

-Are there concerns about ethical or regulatory requirements being met?

Reviewer #1: The revised manuscript adequately addresses the key methodological and conceptual points required for acceptance.

Reviewer #2: The authors addressed the suggestions and made the necessary adjustments. The requested clarifications were incorporated well, and the description of the procedures is now clearer. Overall, the changes improve the transparency of the work.

**Results**

-Does the analysis presented match the analysis plan?

-Are the results clearly and completely presented?

-Are the figures (Tables, Images) of sufficient quality for clarity?

Reviewer #1: The results are clearly and comprehensively presented.

Reviewer #2: I only noted a couple of formatting issues: at line 61 there is a double space in 'Triatoma virus', and at line 112 the spacing between words should be checked ('24tissue').

**Conclusions**

-Are the conclusions supported by the data presented?

-Are the limitations of analysis clearly described?

-Do the authors discuss how these data can be helpful to advance our understanding of the topic under study?

-Is public health relevance addressed?

Reviewer #1: The conclusions are well supported by the data presented and accurately reflect the scope and limitations of the analyses.

Reviewer #2: The Conclusions section is clear and appropriate as presented. I have no further comments.

**Editorial and Data Presentation Modifications?**

Reviewer #1: Needles

Reviewer #2: line 61: double space in "Triatoma virus"

line 112: check spacing between words in the document... "24tissue"

**Summary and General Comments**

Reviewer #1: This represents a valuable contribution to the growing literature on vector-associated viromes and provides a useful baseline for future ecological, evolutionary, and functional investigations.

Reviewer #2: Overall, the manuscript is clearly written and the study is well executed. The topic is relevant, and the results contribute useful information to the field. While there are minor formatting issues that should be corrected, the overall quality of the work is solid and the conclusions are supported by the data.

PLOS authors have the option to publish the peer review history of their article (what does this mean? ). If published, this will include your full peer review and any attached files.

**Do you want your identity to be public for this peer review?** For information about this choice, including consent withdrawal, please see our Privacy Policy .

Reviewer #1: No

Reviewer #2: No

---

## [Editor Report · Acceptance letter]

Dear Assoc. Prof. Novakova,

We are delighted to inform you that your manuscript, "Viral metagenome characterization reveals species-specific virome profiles in Triatominae populations from the southern United States," has been formally accepted for publication in PLOS Neglected Tropical Diseases.

Best regards,

Shaden Kamhawi

co-Editor-in-Chief

Paul Brindley

co-Editor-in-Chief
